# UniBias: Unveiling and Mitigating LLM Bias through Internal Attention and FFN Manipulation

**Hanzhang Zhou**[1,2], **Zijian Feng**[1,2], **Zixiao Zhu**[1,2], **Junlang Qian**[1], **Kezhi Mao**[1,2]
[1]Nanyang Technological University    [2]Singapore-ETH Centre
{hanzhang001, feng0119, zixiao001, junlang001}@e.ntu.edu.sg
ekzmao@ntu.edu.sg

## Abstract

Large language models (LLMs) have demonstrated impressive capabilities in various tasks using the in-context learning (ICL) paradigm. However, their effectiveness is often compromised by inherent bias, leading to prompt brittleness—sensitivity to design settings such as example selection, order, and prompt formatting. Previous studies have addressed LLM bias through external adjustment of model outputs, but the internal mechanisms that lead to such bias remain unexplored. Our work delves into these mechanisms, particularly investigating how feedforward neural networks (FFNs) and attention heads result in the bias of LLMs. By Interpreting the contribution of individual FFN vectors and attention heads, we identify the biased LLM components that skew LLMs' prediction toward specific labels. To mitigate these biases, we introduce UniBias, an inference-only method that effectively identifies and eliminates biased FFN vectors and attention heads. Extensive experiments across 12 NLP datasets demonstrate that UniBias significantly enhances ICL performance and alleviates prompt brittleness of LLMs. The code is available at https://github.com/hzzhou01/UniBias.

## 1 Introduction

Large language models (LLMs) have shown exceptional capabilities in various natural language processing (NLP) tasks, employing the in-context learning (ICL) paradigm. This paradigm conditions LLMs on a context prompt comprising of a few example-label pairs [Brown et al., 2020, Wei et al., 2022, Dong et al., 2023, Zhou et al., 2024].

Despite their impressive performance, LLMs are prone to prompt brittleness, characterized by high sensitivity to the choice [Zhao et al., 2021] and order [Lu et al., 2022] of examples, and prompt formatting [Min et al., 2022], as demonstrated in Figure 1. Such prompt brittleness is found to be arise from the bias in LLMs towards predicting certain answers [Zhao et al., 2021]. The presence of the LLM bias undermines the robustness and adaptability of LLMs in diverse applications.

Extensive research has focused on identifying factors that lead to LLM bias and strategies for mitigation. For instance, vanilla label bias [Fei et al., 2023] and recency bias [Zhao et al., 2021] demonstrate the LLM's inherent non-contextual preference for certain labels and contextual preference for specific positions, respectively. Additionally, several calibration methods [Fei et al., 2023, Han et al., 2023, Zhao et al., 2021] are proposed to counteract the bias by adjusting decision boundaries of model output probabilities. However, these approaches are derived from *external* observations or adjustments of LLM outputs, leaving **the *internal* mechanisms within LLMs that cause such bias poorly understood**.

In this work, we investigate the internal mechanism of LLM bias, specifically how feedforward neural networks (FFNs) and attention heads contribute to such bias. Building on findings in mechanistic

38th Conference on Neural Information Processing Systems (NeurIPS 2024).

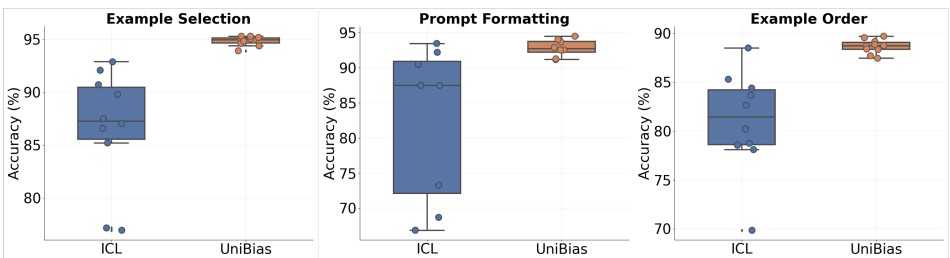

Figure 1: illustrates the prompt brittleness of ICL and the effectiveness of our method in mitigating this issue. Experiments are conducted in one-shot setting, using SST2 [Socher et al., 2013] dataset for experiments on example selection and prompt formatting and AGnews [Zhang et al., 2015] dataset for example order experiment due to more diverse combination of orders.

interpretability [Elhage et al., 2021, Dar et al., 2023], we assess the contribution of individual attention heads and FFN vectors[1] to label predictions in LLMs. By identifying FFN vectors and attention heads that convey biased influences towards label prediction, we reveal the internal mechanisms behind several key bias factors, including vanilla label bias [Fei et al., 2023], recency bias [Zhao et al., 2021], and selection bias [Zheng et al., 2023]. For instance, our analysis of FFN vectors without input context demonstrates that their cumulative impact biases the LLM towards specific labels, indicating a non-contextual preference for certain labels, i.e., vanilla label bias. We elaborate on the background of mechanistic interpretability in Section 2.1 and present our findings on the internal mechanisms of LLM biases in next section.

Given our findings that various bias factors stem from the biased behaviors of attention heads and FFN vectors, we are prompted to ask: Can we identify the biased components of LLMs and mitigate their detrimental impact on label prediction? Motivated by this intuition, we propose **UniBias**, an inference-only method designed to identify and eliminate biased FFN vectors and attention heads in LLMs. Specifically, we begin by projecting each FFN vector and attention head into the vocabulary space to interpret the information conveyed by their outputs. We then detect biased components based on three criteria we defined: the relatedness criterion, the bias criterion, and the low variance criterion. After identification, we mitigate their impact by masking these biased components. Extensive experimental results demonstrate that LLMs, from which biased components have been removed, consistently outperform their original counterparts by a significant margin. Further, as illustrated in Figure 1, our method significantly improves both the performance and robustness of ICL with perturbations of various design settings.

The contributions of our work are summarized as follows:

- In contrast to existing works based on external adjustments of LLM outputs, we mitigate LLM bias through manipulation of LLM internal structure. This novel perspective potentially offers a new direction for the field. Moreover, our method demonstrate an effective way to manipulate internal structures of LLMs.

- We conduct a thorough investigation of the internal mechanisms underlying biases in LLMs, revealing the inner causes of these biases.

- Extensive experiments across 12 NLP datasets demonstrate that, by removing the biased components, our UniBias method significantly enhances ICL performance and achieve state-of-the-art results. Additionally, it effectively addresses the issue of prompt brittleness.

## 2 Internal Mechanisms Causing the Bias of LLMs

This section reveals the internal mechanisms within LLMs that lead to various bias factors.

---

[1]FFN vector refers to the value vector in the second weight matrix of the FFN layer. We elaborate on this in Section 2.1

## 2.1 Background

The theoretical background of this work is based on research on mechanistic interpretability [Elhage et al., 2021, Wang et al., 2022, Geva et al., 2021], which aims to explain the internal processes in language models (LMs), facilitating the interpretation of the contributions of individual model components to the final prediction.

We are focusing on decoder-only LMs in this paper. They are composed by a sequence of transformer layers, each composed of a multi-head self-attention layer and an feedforward neural network layer. The background knowledge for interpreting the contribution of each FFN vector and attention head to the models' prediction are demonstrated as follows.

**The Residual Stream** We interpret Transformers following the view of residual stream [Elhage et al., 2021, Dar et al., 2023]. Due to the residdual connection of Transformers, each layer takes a hidden state as input, and adds information obtained by its attention layer and FFN layer to the hidden state through residual connection. In this sence, the hidden state is a residual stream passed along layers, and each attention layer and FFN layer contribute to the final prediction by adding information to the residual stream.

**Attention Heads** Following Elhage et al. [2021], Dar et al. [2023], the output of each attention layer of LM can be computed as the sum of all its attention heads. Specifically, for $l$-th layer, the input is $X^l \in \mathbb{R}^{N \times d}$, and the attention layer is parameterized by four matrices $W_Q^l, W_K^l, W_V^l, W_O^l \in \mathbb{R}^{d \times d}$. The columns of each projection matrix and the rows of the output matrix can be split into $H$ parts: $W_Q^{\ell,j}, W_K^{\ell,j}, W_V^{\ell,j} \in \mathbb{R}^{d \times \frac{d}{H}}$ and $W_O^{\ell,j} \in \mathbb{R}^{\frac{d}{H} \times d}$, where $H$ is the number of attention heads. We then find that:

$$\text{Att}^\ell(X^\ell) = \textbf{Concat} \left[ A^{\ell,1} X^\ell W_V^{\ell,1}, A^{\ell,2} X^\ell W_V^{\ell,2}, \dots, A^{\ell,H} X^\ell W_V^{\ell,H} \right] W_O^\ell = \sum_{j=1}^{H} A^{\ell,j}(X^\ell W_V^{\ell,j}) W_O^{\ell,j}$$

where $A^{\ell,j} = \text{softmax} \left( \frac{(X^\ell W_Q^{\ell,j})(X^\ell W_K^{\ell,j})^T}{\sqrt{d/H}} + M^{\ell,j} \right)$, $M^{\ell,j}$ is the attention mask. Therefore, the output of an attention layer is equivalent to computing attention heads independently, multiplying each by its own output matrix, and adding them into the residual stream of the LM.

**FFN** In line with Geva et al. [2021, 2022], transformer FFN layers can be cast as linear combination of vectors. Specifically, for an input vector $\mathbf{x}^\ell \in \mathbb{R}^d$, FFN parameter matrices $\mathbf{K}^\ell, \mathbf{V}^\ell \in \mathbb{R}^{d_m \times d}$, the FFN output can be derived as:

$$\text{FFN}^\ell(\mathbf{x}^\ell) = f(\mathbf{x}^\ell \mathbf{K}^{\ell T}) \mathbf{V}^\ell = \sum_{i=1}^{d_m} f(\mathbf{x}^\ell \cdot \mathbf{k}_i^\ell) \mathbf{v}_i^\ell = \sum_{i=1}^{d_m} m_i^\ell \mathbf{v}_i^\ell$$

where $f$ is the activation function, $i$ is the index of the vector. Then, the FFN layer can be viewed as a linear combination of vectors: the multiplication of $\mathbf{x}^\ell$ and the key vector $\mathbf{k}_i$ produces the coefficient $m_i^\ell$ that weights the corresponding value vector $\mathbf{v}_i$.

**Logit Lens** The logit lens [Nostalgebraist, 2020] is a technique that directly decode hidden states into the vocabulary space using the unembedding matrix of the LLM for interpretation. This approach has been validated in various studies as an efficient method for interpreting the weight matrix or hidden states of LLMs [Dar et al., 2023, Hanna et al., 2023, Feng et al., 2024, Yu et al., 2023, Geva et al., 2021].

In summary, each attention layer and FFN layer contribute to the final prediction by adding their output hidden states to the residual stream. These outputs can be viewed as the sum of their respective attention heads and FFN vectors. Each attention head or FFN vector's output can be interpreted through the logit lens.

## 2.2 Internal Mechanisms of Bias Factors

We delve into the mechanisms behind several bias factors, analyzing the contributions of attention heads and FFN vectors to the biased predictions in LLMs. We explore vanilla label bias, position bias, and selection bias using the Llama-2 7B model [Touvron et al., 2023].

**Vanilla Label Bias** The vanilla label bias [Fei et al., 2023], also known as common token bias [Zhao et al., 2021], is the inherent, uncontextual preference of the model towards predicting certain label names. Given the contextual nature of attention layers, our investigation focuses on the FFN layers, where we identified a corresponding uncontextual preference. Specifically, by projecting the FFN value vectors into the vocabulary space, we compute the logits for various label names for each FFN vector. Utilizing the residual stream insight, we then aggregate these logits for all FFN vectors whose label logits rank within the top 10 over the vocabulary, reflecting uncontextual influences of FFN vectors that are effective in label prediction. This process yields what we term *uncontextual accumulated FFN logits*, revealing the intrinsic bias of the LLM towards predicting label names without the influence of input.

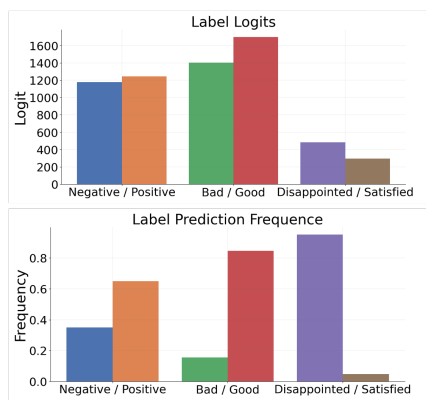

Figure 2: Unveiling vanilla label bias by uncontextual accumulated FFN logits.

Figure 2 illustrates the accumulated uncontextual FFN logits across different label names in the sentiment analysis task, alongside their corresponding zero-shot prediction frequencies on the SST-2 dataset. For example, the label name 'positive' exhibits higher uncontextual accumulated FFN logits compared to 'negative,' leading to a higher frequency of 'positive' predictions. Additionally, when comparing the labels 'good' and 'bad', the difference in their uncontextual accumulated FFN logits is more pronounced than that between 'positive' and 'negative,' resulting in a larger discrepancy in prediction frequency. Conversely, the accumulated logits for the labels 'satisfied' and 'disappointed' show a reverse trend relative to 'positive' and 'negative', which results in a corresponding reverse trend in their prediction frequency ratios.

**Recency Bias** Recency bias refers to the tendency of LLMs to favor the label of the example at the end of the prompt [Zhao et al., 2021]. By examining the behavior of attention heads within LLMs, we observe that specific heads consistently prioritize the example at the end of the prompt, providing an internal perspective on the origin of recency bias.

We identify the biased attention head using the method introduced in Section 3. We compare the behaviors of a biased attention head (layer 16, head 29) and an unbiased attention head (layer 16, head 19) in terms of the attention weight as-

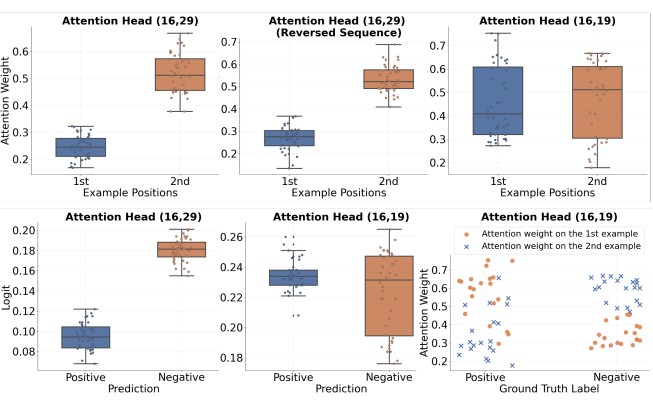

Figure 3: The internal mechanism of the recency bias.

signed to examples at different positions and the label logits of the corresponding attention head's output. Specifically, we use the SST-2 dataset, including one positive and one negative example in the prompt, and test with 40 samples, evenly split between positive and negative examples. More experimental details are provided in Appendix A.

Experimental results in Figure 3 reveal that the biased attention head (layer 16, head 29) consistently assigns significantly larger attention weights to the final example, irrespective of the ground truth labels of the test samples. This bias persists even when the sequence of examples is reversed, as shown in the second subfigure, indicating a biased preference of this attention head for the last example in the prompt. Furthermore, the biased attention weight assignment leads to biased logits, as shown in the third subfigure. In contrast, the unbiased attention head (layer 16, head 19) assigns very close averaged attention weights to both examples in the prompt. Interestingly, we observe that this unbiased head generally assigns larger weights to the example whose label matches the ground

truth label of the test sample, resulting in 35 out of 40 samples being correctly classified based on this pattern by this single attention head. The preference shown by specific attention heads for the example at the end of the prompt reveals the internal mechanism of recency bias.

**Selection Bias** The selection bias refers that LLMs prefer to select specific option ID (like "Option A") as answers for multiple choice questions [Zheng et al., 2023]. We have identified both FFN vectors and attention heads that consistently favor a specific option regardless of the ground truth label of the test sample, revealing the internal mechanism of selection bias.

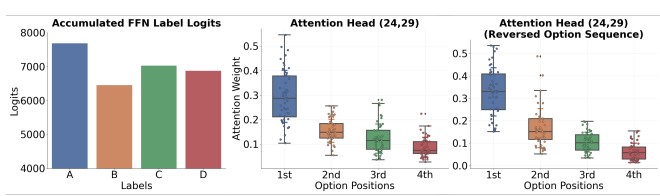

Figure 4: The internal mechanism of the selection bias.

We evaluate the Llama-2 7B model on the ARC dataset, which contains four options (A, B, C, D). We use a zero-shot setting to avoid the influence of position bias from multiple examples. More details are provided in Appendix A. Experimental results are illustrated in Figure 4. Firstly, we observe that the LLM exhibits a vanilla label bias favoring option "A", as shown in the first subfigure. Additionally, we identify a biased attention head that demonstrates a position bias consistently favoring the first option regardless of the ground truth labels of the test samples (second subfigure) or changes in the sequence of options (third subfigure). Since option A is usually the first option, these two biases both lead to the LLM's preference for option A.

## 3   Methodology

In the previous section, we unveil that **various bias factors are stem from the biased behaviors of attention heads and FFN vectors**. Naturally, we pose the question: *Can we identify the biased components of LLMs and mitigate their impact on label prediction?* Therefore, we propose our **UniBias** method to **Un**veil and **m**itigate LLMs' label **Bias** through internal attention and FFN manipulation. Notably, our method is proposed for decoder-only LLMs.

### 3.1   Biased FFN Vectors Identification

Identifying biased FFN vectors in LLMs hinges on whether the contribution of each FFN vector is independent and interpretable. As discussed in Section 2.1, the output of an FFN layer can be cast as a linear combination of FFN vectors. Each FFN vector contributes to the final prediction by adding information encoded in its value vector, $\mathbf{v}_i^\ell$, weighted by its corresponding coefficient, $m_i^\ell$. This information within $\mathbf{v}_i^\ell$ can be interpreted through the logit lens, enabling us to interpret it as a distribution of logits across the vocabulary space.

How to identify an FFN vector as biased? we assess whether it consistently introduces a biased preference towards specific labels into the residual stream, regardless of variations in the test samples. Such consistent biases can skew the LLM's predictions. We introduce the following criteria to detect biased components in LLMs, which are also applicable for identifying biased attention heads:

- **Relatedness Criterion**: The information introduced by the FFN vector (or attention head) should closely relate to label prediction.

- **Biased Criterion**: The information contributed to the residual stream by the FFN vector (or attention head) exhibits a biased distribution, favoring certain labels over others.

- **Low Variance Criterion**: The label prediction information added by the FFN vector (or attention head) to the residual stream is almost identical across a set of test samples with different labels, i.e., exhibits very small variance.

The third criterion is key to identifying biased FFN vectors (or attention heads), as consistently low variance indicates that the FFN vector is not adequately responsive to varying inputs. Combined with the second criterion, this suggests a bias towards certain predictions regardless of the input's contextual differences.

To examine these criteria, we interpret the information contributed by each FFN vector, i.e., $m\mathbf{v}$. For simplicity, we omit the layer number $\ell$ and FFN index $i$. Since the FFN value vector $\mathbf{v}$ is fixed, changes in the FFN coefficient $m$ across different samples reflect the change in information brought by the FFN vector. We interpret this information by projecting each FFN value vector into the vocabulary space and analyzing the logit distribution over label tokens, termed *label logits*.

Specifically, given an FFN value vector $\mathbf{v} \in \mathbb{R}^d$, the unembedding matrix $E \in \mathbb{R}^{d \times d_e}$, a label token mapping matrix $L \in \mathbb{R}^{N \times d_e}$, where each row is a one-hot vector indicating the token id of the first token of each label name, the label logits $\mathbf{g}^{(\mathbf{k})} = [g_0^{(k)}, g_1^{(k)}, \ldots, g_{c-1}^{(k)}]^\top$ (where $c$ is the class number) corresponding to the FFN value vector $\mathbf{v}$ of $k$-th sample can be obtained by:

$$\mathbf{g} = \mathbf{v} \cdot E \cdot L^\top$$

We use $p$ unlabeled samples from the task to assess the three criteria we defined. The coefficients and label logits of an FFN vector for these samples are denoted as $\mathbf{m} = [m_0, m_1, \ldots, m_{p-1}]$ and $\mathbf{G} = [\mathbf{g}^{(0)}, \mathbf{g}^{(1)}, \ldots, \mathbf{g}^{(p-1)}]^\top \in \mathbb{R}^{p \times c}$, respectively. An FFN vector is considered biased if it meets the following conditions, each corresponding to one of the three criteria we defined:

$$
\begin{cases}
\dfrac{1}{p} \sum_{k=0}^{p-1} \mathrm{Sum}\left(\mathbf{G}_{k,:}\right) = \dfrac{1}{p} \sum_{k=0}^{p-1} \mathrm{Sum}\left(\mathbf{g}^{(\mathbf{k})}\right) = \dfrac{1}{p} \sum_{k=0}^{p-1} \sum_{j=0}^{c-1} g_j^{(k)} > th_{FFN}^1 & (1) \\[4mm]
\dfrac{1}{p} \sum_{k=0}^{p-1} \mathrm{Bias}\left(\mathbf{G}_{k,:}\right) = \dfrac{1}{p} \sum_{k=0}^{p-1} \mathrm{Bias}\left(\mathbf{g}^{(\mathbf{k})}\right) = \dfrac{1}{p}\dfrac{1}{c} \sum_{k=0}^{p-1} \sum_{j=0}^{c-1} \left(g_j^{(k)} - \mu(\mathbf{g}^{(\mathbf{k})})\right) > th_{FFN}^2 & (2) \\[4mm]
CV\left(\mathbf{m}\right) = \dfrac{\sigma(\mathbf{m})}{\mu(\mathbf{m})} = \dfrac{\sqrt{\frac{1}{p} \sum_{j=0}^{p-1} \left(m_k - \mu(\mathbf{m})\right)^2}}{\frac{1}{p} \sum_{k=0}^{p-1} m_k} < th_{FFN}^3 & (3)
\end{cases}
$$

where $\mu(\mathbf{g}^{(k)}) = \frac{1}{c} \sum_{j=0}^{c-1} g_j^{(k)}$, $\mu(\mathbf{m}) = \frac{1}{p} \sum_{k=0}^{p-1} m_k$. The thresholds $th_{FFN}^1, th_{FFN}^2, th_{FFN}^3$ are set by grid search, which is elaborated in Section 3.4

The first equation corresponds to the relatedness criterion, measured by the sum of label logits. A higher sum indicates that the information introduced by the FFN vector is more relevant to label prediction. The second equation relates to the bias criterion, quantified by the deviation of the average logit for each label from the overall average logit across all labels. Ideally, for a set of test samples with different labels, the average logits for each label should be relatively balanced. A greater deviation from each label's average compared to the overall average across all labels indicates a more biased distribution. The third equation addresses the low variance criterion, measured by the coefficient of variation (CV) of the FFN vector coefficients across different samples. The CV, calculated as the standard deviation normalized by the mean, indicates whether the label prediction information added by the FFN vector remains almost the same across different samples.

## 3.2 Biased Attention Heads Identification

The identification of biased attention heads closely resembles the process of identifying biased FFN vectors. As discussed in Section 2.1, each attention head's contribution to the final prediction is independent and interpretable. Therefore, we project the output hidden states of each attention head into the vocabulary space to interpret the information they contribute.

To identify biased attention heads, we use the same three criteria introduced for identifying biased FFN vectors. To apply these criteria, we project the output hidden states from each attention head into the vocabulary space and analyze their label logits as the information contributes to label prediction. The output from each attention head consists of hidden states generated for every token in the sequence. For our analysis, we specifically use the hidden state of the last token preceding the prediction of label names, interpreting it as the most direct contribution of the attention head to the prediction, given the autoregressive nature of LLMs.

Specifically, to obtain the label logits for an attention head, consider the output hidden states $H \in \mathbb{R}^{N \times d}$ of this head, the unembedding matrix $E \in \mathbb{R}^{d \times d_e}$, and the label token mapping matrix $L \in \mathbb{R}^{N \times d_e}$. Given the token position $p_{\text{label}} \in \{0, 1, \ldots, N-1\}$, which indicates the index of the first token of the predicted label names, the label logits $\mathbf{a}^{(k)} = [a_1^{(k)}, a_2^{(k)}, \ldots, a_c^{(k)}]^\top$ of the attention head for the $k$-th sample are derived by:

$$\mathbf{a}^{(k)} = H_{(p_{\text{label}}-1),:} \cdot E \cdot L^\top.$$

we employ the same $p$ unlabeled samples from the task to assess the criteria for identifying baised attention head. The label logits for these samples are formed as $A = [\mathbf{a}^{(0)}, \mathbf{a}^{(2)}, \ldots, \mathbf{a}^{(m-1)}]^{\top} \in \mathbb{R}^{m \times c}$. An attention head is considered biased if it meets the following conditions:

$$
\begin{cases}
\frac{1}{p} \sum_{k=0}^{p-1} \text{Sum}\left(A_{k,:}\right) = \frac{1}{p} \sum_{k=0}^{p-1} \text{Sum}\left(\mathbf{a}^{(k)}\right) = \frac{1}{p} \sum_{k=0}^{p-1} \sum_{j=1}^{c} \mathbf{a}_j^{(k)} > th_{Att}^1 \\[2ex]
\frac{1}{p} \sum_{k=0}^{p-1} \text{Bias}\left(A_{k,:}\right) = \frac{1}{p} \sum_{k=0}^{p-1} \text{Bias}\left(\mathbf{a}^{(k)}\right) = \frac{1}{p} \frac{1}{c} \sum_{k=0}^{p-1} \sum_{j=0}^{c-1} \left(a_j^{(k)} - \mu(\mathbf{a}^{(k)})\right) > th_{Att}^2 \\[2ex]
\sum_{j=0}^{c-1} w_j \cdot CV\left(A_{:,j}\right) = w_j \cdot \frac{\sigma(A_{:,j})}{\mu(A_{:,j})} < th_{Att}^3
\end{cases}
$$

where $w_j = \sum_{j=0}^{c-1} \frac{\mu(A_{:,j})}{\sum \mu(A_{:,j})}, \mu(A_{:,j}) = \frac{1}{p} \sum_{k=0}^{p-1} A_{i,j}, \sigma(A_{:,j}) = \sqrt{\frac{1}{p} \sum_{k=0}^{p-1} (A_{i,j} - \mu(A_{:,j}))^2}$.
The functions of the first two criteria are identical to those for biased FFN vector identification. The third function is the weighted sum of the coefficient variance of each label across test samples. The thresholds for biased attention head identification are also derived by grid search.

### 3.3 Biased FFN Vectors and Attention Heads Manipulation

After identifying the biased components of the LLM, we eliminate their influence by masking these biased FFN vectors and attention heads. Specifically, we create masks for the attention heads in each attention layer and reset the coefficient of the biased FFN vector and biased attention head mask.

### 3.4 Grid Searching

Specifically, we utilize a small subset of training data as a support set, with 20 samples for each class. We then grid search all combinations of threshold values and select the combination that results in the most balanced distribution of average label logits. Specifically, let $\mathbf{T}$ represents the set of threshold combinations, and $P(t)$ denote the average label logits for a threshold combination $t \in \mathbf{T}$, we aim to find the combination $t^*$ that minimizes the bias of label logits: $t^* = \arg\min_{t \in \mathbf{T}} \text{Bias}(\mathbf{P}(t))$.

It is noteworthy that although there are multiple combinations of thresholds, they usually result in a few set of different biased components. For example, for a grid search of thresholds of FFN vectors with 80 combinations, it only result in 4 different sets of biased FFN vectors that need to be examined with the support set on the SST-2 dataset. Additionally, during the inference stage of evaluating test samples, the computation time of the UniBias method is completely identical to that of the original LLMs.

Additionally, the support set can be replaced with unlabeled samples, using approximately twice the number of unlabeled samples compared to labeled ones. For further details, please see Appendix F.

## 4 Experiments

In this section, we aims to investigate a few research questions (RQ). **RQ 1**: After eliminating biased components from LLMs, does the ICL performance improve compared to the original LLM? Additionally, how does our UniBias method compare to existing calibration methods? **RQ 2**: Given that ICL suffers from prompt brittleness, can our UniBias method contribute to more robust ICL performance? **RQ 3**: Are there any observable patterns of biased FFN vectors and attention heads within and across tasks? **RQ 4**: What is the performance of LLMs after eliminating only the biased FFN vectors and only the biased attention heads, respectively? **RQ 5**: What is the impact of support set size on the performance of the UniBias method?

### 4.1 Experimental Setup

**Datasets** We evaluate our UniBias method on 12 diverse natural language processing datasets across various tasks, including sentiment analysis, topic classification, natural language inference, reasoning, and word disambiguation. Statistics and details about the datasets can be found in Table 4 in Appendix.

Table 1: Comparison of one-shot ICL performance for different methods across datasets using Llama-2 7b and Llama-2 13b models. The mean and standard deviation are reported for five repetitions with different ICL examples.

| Dataset | Llama-2 7b | | | | | Llama-2 13b | | | | |
|---|---|---|---|---|---|---|---|---|---|---|
| Method | ICL | CC | DC | PC | UniBias | ICL | CC | DC | PC | UniBias |
| SST-2 | $87.22_{6.03}$ | $92.24_{3.39}$ | $94.15_{1.22}$ | $93.90_{1.54}$ | $\mathbf{94.54}_{0.62}$ | $93.90_{1.79}$ | $95.25_{0.93}$ | $95.37_{0.70}$ | $94.56_{1.71}$ | $\mathbf{95.46}_{0.52}$ |
| MNLI | $53.83_{2.22}$ | $53.36_{3.16}$ | $52.19_{2.55}$ | $45.38_{5.01}$ | $\mathbf{54.97}_{0.88}$ | $62.43_{1.49}$ | $63.89_{0.81}$ | $61.86_{1.23}$ | $57.47_{3.53}$ | $\mathbf{64.65}_{2.73}$ |
| WiC | $50.00_{0.16}$ | $52.19_{2.00}$ | $52.40_{1.69}$ | $\mathbf{57.11}_{2.49}$ | $53.71_{1.16}$ | $54.48_{3.19}$ | $50.63_{1.73}$ | $49.72_{0.30}$ | $55.67_{1.67}$ | $\mathbf{57.93}_{1.70}$ |
| COPA | $67.60_{2.30}$ | $67.80_{2.17}$ | $60.40_{2.79}$ | $67.80_{3.70}$ | $\mathbf{69.00}_{2.74}$ | $67.50_{10.40}$ | $75.20_{7.80}$ | $71.00_{8.80}$ | $76.80_{6.30}$ | $\mathbf{83.20}_{2.70}$ |
| CR | $91.54_{0.39}$ | $92.13_{0.40}$ | $92.61_{0.44}$ | $91.97_{0.35}$ | $\mathbf{92.61}_{0.11}$ | $91.01_{1.30}$ | $92.13_{0.88}$ | $92.23_{0.76}$ | $91.65_{0.64}$ | $\mathbf{92.34}_{0.74}$ |
| AGNews | $85.59_{1.87}$ | $83.54_{1.96}$ | $\mathbf{89.08}_{0.86}$ | $86.81_{2.92}$ | $88.29_{1.24}$ | $89.14_{0.44}$ | $88.23_{1.14}$ | $\mathbf{89.34}_{0.61}$ | $86.03_{0.65}$ | $88.68_{0.43}$ |
| MR | $89.37_{1.83}$ | $91.77_{1.42}$ | $\mathbf{92.35}_{0.23}$ | $91.39_{1.65}$ | $92.19_{0.37}$ | $90.10_{2.10}$ | $\mathbf{93.20}_{0.57}$ | $93.00_{0.52}$ | $92.80_{0.86}$ | $92.23_{1.12}$ |
| RTE | $66.21_{7.30}$ | $64.33_{3.68}$ | $65.49_{2.09}$ | $62.59_{4.71}$ | $\mathbf{67.65}_{6.44}$ | $76.10_{4.73}$ | $71.99_{5.02}$ | $66.21_{1.09}$ | $75.31_{2.90}$ | $\mathbf{78.23}_{2.13}$ |
| SST-5 | $46.97_{0.87}$ | $51.36_{1.69}$ | $51.92_{1.77}$ | $\mathbf{55.41}_{1.51}$ | $53.79_{1.46}$ | $51.03_{1.25}$ | $47.20_{1.69}$ | $48.98_{2.11}$ | $\mathbf{53.63}_{0.95}$ | $51.80_{1.00}$ |
| TREC | $72.92_{12.42}$ | $76.44_{3.21}$ | $77.16_{3.94}$ | $74.92_{5.78}$ | $\mathbf{80.80}_{3.17}$ | $74.70_{12.10}$ | $\mathbf{83.80}_{3.86}$ | $80.50_{9.07}$ | $81.85_{9.53}$ | $81.25_{6.86}$ |
| ARC | $51.90_{0.60}$ | $\mathbf{53.10}_{0.40}$ | $53.00_{0.60}$ | $40.40_{0.50}$ | $\mathbf{53.10}_{0.60}$ | $66.54_{0.33}$ | $64.33_{0.99}$ | $64.88_{0.59}$ | $59.47_{1.07}$ | $\mathbf{66.81}_{0.37}$ |
| MMLU | $41.73_{2.25}$ | $43.72_{0.97}$ | $43.57_{1.38}$ | $34.12_{3.41}$ | $\mathbf{44.83}_{0.24}$ | $53.53_{1.55}$ | $50.84_{1.57}$ | $51.81_{1.24}$ | $45.50_{1.65}$ | $\mathbf{53.55}_{1.05}$ |
| Avg. | 67.07 | 68.49 | 68.70 | 66.81 | **70.46** | 72.54 | 73.06 | 72.08 | 72.56 | **75.51** |

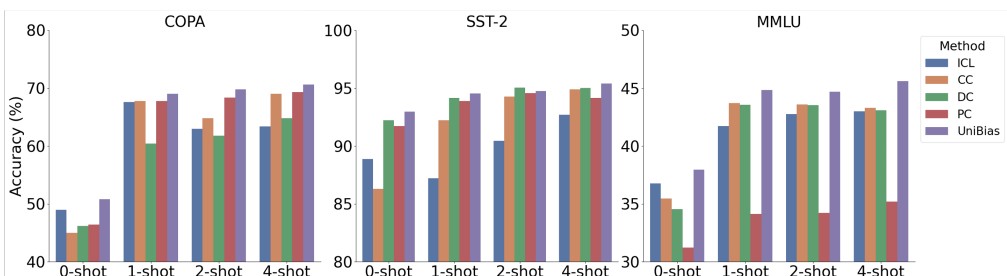

Figure 5: The performance comparison under different numbers of ICL shots using Llama-2-7b.

**Baselines** In addition to the standard ICL, we compare our proposed UniBias with state-of-the-art LLM debiasing and calibration baselines, including **Contextual Calibration (CC)** [Zhao et al., 2021], **Domain-Context Calibration (DC)** [Fei et al., 2023], and **Prototypical Calibration (PC)** [Han et al., 2023]. We reproduce all baselines strictly follows the authors' instructions and recommendations to ensure a fair comparison.

**Models and implementation details** We evaluate our method using a range of LLMs, including Llama-2 7b, Llama-2 13b [Touvron et al., 2023], GPT-J [Wang and Komatsuzaki, 2021] and GPT2-XL [Radford et al., 2019]. For all experiments, unless stated otherwise, we use 1-shot ICL setting, i.e. one example per class, and repeat five times under different random seeds. We use $k = 20$ sampes per class as the support set to obtain all threshold values by grid searching, as mentioned in the method section. The prompt template and more implementation details are specified in Appendix A.

### 4.2 Main Experiments

Table 1 presents the performance of various datasets and model sizes under the 1-shot setting. Our proposed UniBias method consistently achieves the highest accuracies in most cases. In terms of overall average accuracy, UniBias improves upon the standard ICL by a substantial margin of 3.39% and exceeds the state-of-the-art (SOTA) DC by 1.76% using Llama-2 7b. With Llama-2 13b, UniBias surpasses the standard ICL and the SOTA CC by 2.97% and 2.45%, respectively. Figure 5 further illustrates the results under zero-shot and various few-shot settings for COPA, SST2, and MMLU. Additionally, to demonstrate the effectiveness of our method across different large language models, we present results for GPT-J and GPT2-XL in Figure 7 of Appendix C. Our proposed UniBias consistently surpasses other baselines in all scenarios, underscoring its effectiveness.

In response to **RQ 1**, UniBias not only enhances the performance of original LLMs but also outperforms existing methods. We attribute this success to its internal analysis and bias mitigation techniques, which leverage FFNs and attentions, unlike other methods that rely solely on external observations.

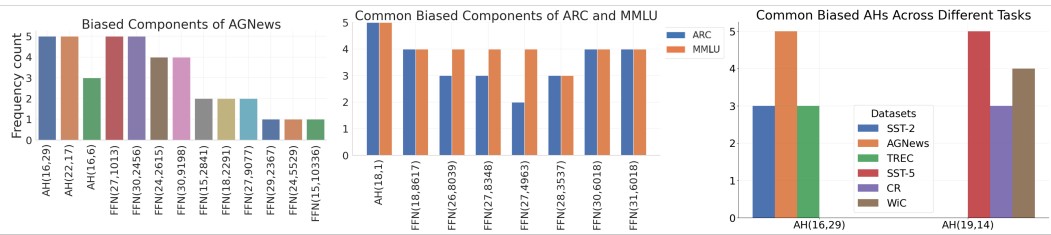

Figure 6: Analysis of biased attention heads (AHs) and FFN vectors (FFNs). The frequency count of biased LLM components across five repeat experiments with different example selections is reported.

Table 2: Experiments on eliminating common biased components. Attention heads that are frequently identified as biased are removed from the original Llama-2 7b model.

|  | SST2 | MMLU | COPA | RTE | MR | Trec | Avg. |
|---|---|---|---|---|---|---|---|
| ICL | $87.22_{6.03}$ | $41.73_{2.25}$ | $67.60_{2.30}$ | $66.21_{7.30}$ | $89.37_{1.83}$ | $72.92_{12.42}$ | 70.84 |
| Unibias | $\mathbf{94.54}_{0.62}$ | $\mathbf{44.83}_{0.24}$ | $\mathbf{69.00}_{2.74}$ | $\mathbf{67.65}_{6.44}$ | $92.19_{0.37}$ | $\mathbf{80.80}_{3.17}$ | $\mathbf{74.84}$ |
| Eliminating Common Biased Components | $94.32_{0.60}$ | $44.20_{1.14}$ | $68.00_{2.87}$ | $67.37_{4.60}$ | $\mathbf{92.43}_{0.09}$ | $77.60_{4.75}$ | 73.98 |

## 4.3 Alleviating Prompt Brittleness

Existing studies have found that LLMs are prone to prompt brittleness, with various factors such as the selection and order of examples, as well as the prompt formatting. To address **RQ 2**, we simulate these brittle scenarios by choosing different demonstration samples, using different prompt formats, and changing the example order to observe variations in LLM performance.

Figure 1 presents Llama-2 7b's performance both with and without UniBias. Without UniBias, the standard ICL's performance varies significantly, ranging from 8% to 26%, demonstrating its instability. After applying UniBias, the accuracy remains consistently high and stable, with variations consistently less than 4% under perturbations of various design settings. We provide further theoretical analysis on why UniBias can mitigate prompt brittleness and address various bias factors in Appendix G.

## 4.4 Biased Components Analysis and Common Biased Components Elimination

In response to **RQ3**, we present the frequency counts of identified biased attention heads (AHs) and FFNs under repeated experiments in Figure 6. A large frequency count for an LLM component indicates a higher repeat of being identified as biased in the corresponding dataset. The first subfigure displays the biased components for various example selections, revealing several commonly biased LLM components across different prompts within a single dataset. The second subfigure highlights the common biased components across different datasets (ARC and MMLU) for the reasoning task, indicating that different datasets with similar tasks could share common biased LLM components. The third subfigure demonstrates the presence of common biased LLM components across different tasks.

Experimental results suggest an interesting future direction: we may identify global biased components that would mitigate bias across multiple tasks and diverse prompt design settings. We conduct an preliminary experiment to explore the potential of eliminating common biased components. Specifically, we eliminate attention heads that are frequently identified as biased and apply this setting to diverse tasks, rather than handling each task individually. Experimental results in Table 2 demonstrate that although not as effective as our full Unibias method, eliminating common biased components outperforms the vanilla ICL by a large margin. Experiment details are in Appendix D.

## 4.5 Ablations

We conduct ablation studies to analyze the impact of exclusively eliminating biased AHs or FFNs to address **RQ 4**. Table 3 presents the results of removing only biased FFN vectors (FFN-only) and only

Table 3: Performance comparison of only removing biased FFN vectors (FFN-only), only removing biased attention heads (attention-only), our Unibias method, and the ICL of original LLM.

| Method | SST-2 | MNLI | WiC | COPA | CR | AGNews | MR | RTE | SST-5 | TREC | ARC | MMLU |
|---|---|---|---|---|---|---|---|---|---|---|---|---|
| ICL | 87.22 | 53.83 | 50.00 | 67.60 | 91.54 | 85.59 | 89.37 | 66.21 | 46.97 | 72.92 | 51.90 | 41.73 |
| FFN-only | 94.17 | 54.59 | 50.88 | 69.20 | 92.57 | 85.52 | 91.78 | 67.33 | 47.09 | 73.04 | 51.92 | 42.62 |
| Attention-only | 94.22 | 52.83 | 52.76 | 68.50 | 91.49 | 86.25 | **92.61** | 66.55 | 52.68 | 80.68 | 53.00 | 44.67 |
| UniBias | **94.54** | **54.97** | **53.71** | **69.00** | **92.61** | 88.29 | 92.19 | **67.65** | **53.79** | **80.80** | **53.10** | **44.83** |

biased attention heads (attention-only). Both FFN-only and attention-only methods outperform the standard ICL, demonstrating their effectiveness. When combined as UniBias, the method achieves the best results across most datasets, indicating that the two approaches are complementary.

Additionally, we further conduct experiments to investigate the impact of support set size (**RQ 5**), which is detailed in Appendix E.

## 5 Related Work

**Bias in LLMs**: It is well recognized that LLMs are unstable under various ICL design settings, and this instability arises from biases in LLMs toward predicting certain answers [Zhao et al., 2021, Lu et al., 2022]. To understand these biases, existing studies have identified various bias factors, including recency bias, majority label bias, common token bias [Zhao et al., 2021], and domain label bias [Fei et al., 2023] in classification tasks. More recently, selection bias, which consistently favors specific options in multiple-choice questions, has also been identified [Zheng et al., 2023, Wang et al., 2023b]. To address these biases, several calibration methods have been proposed, including contextual calibration [Zhao et al., 2021], domain-context calibration [Fei et al., 2023], and prototypical calibration [Han et al., 2023]. However, these identified bias factors and calibration methods are derived from external observations or adjustments of LLM outputs, leaving the underlying mechanisms within LLMs that cause such biases poorly understood.

**Prompt Brittleness**: Regarding prompt brittleness, it is demonstrated in the literature that this instability of prompt arises from LLMs' inherent bias towards predicting certain answers [Zhao et al., 2021]. Therefore, current research efforts address the prompt brittleness by mitigating LLMs' bias towards labels [Fei et al., 2023, Han et al., 2023, Zhao et al., 2021].

**Mechanistic Interpretability**: Mechanistic interpretability [Elhage et al., 2021, Wang et al., 2022] aims to explain the internal processes in language models, facilitating the interpretation of the contributions of individual model components to the final prediction. Our work builds on the understanding of the residual stream [Elhage et al., 2021], the logit lens [Nostalgebraist, 2020], and the interpretation of LLM components in the vocabulary space [Dar et al., 2023, Geva et al., 2021].

## 6 Conclusion

In this work, we have deepened the understanding of biases in LLMs by unveiling the internal mechanisms that contribute to various bias factors. Building on this understanding, we proposed our UniBias method to mitigate these biases by identifying and eliminating biased FFN vectors and attention heads, demonstrating an effective way to manipulate the internal structures of LLMs. Extensive experiments show that our UniBias method achieves state-of-the-art performance across 12 NLP datasets and different ICL settings. Additionally, our method successfully alleviates prompt brittleness and enhances the robustness of ICL.

## Acknowledgments

The authors would like to thank Edmond Lo, Lihui Chen, Xiyu Zhang, and the anonymous reviewers for their constructive comments and suggestions. The research was conducted at the Future Resilient Systems at the Singapore-ETH Centre, which was established collaboratively between ETH Zurich and the National Research Foundation Singapore. This research is supported by the National Research Foundation Singapore (NRF) under its Campus for Research Excellence and Technological Enterprise (CREATE) programme.

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

# A  Experimental Details

## A.1  Datasets

We evaluate our Unibias method using 12 diverse natural language processing datasets across tasks such as sentiment analysis, topic classification, reasoning, natural language inference, and word disambiguation, as presented in Table 4. In our experiments, we utilize $k$ (where $k = 0, 1, 2, 4$) training samples per class as prompt examples for $k$-shot ICL. For testing, we randomly select 2000 samples for MMLU and 3000 samples for MNLI and MR, while employing the original testing sets for other datasets. Detailed dataset statistics are available in Table 4.

Table 4: Detailed Dataset information

| Dataset | # Classes | # Testing Size |
|---|---|---|
| *Sentiment classification* | | |
| SST2 [Socher et al., 2013] | 2 | 872 |
| SST-5 [Socher et al., 2013] | 5 | 2210 |
| MR [Pang and Lee, 2005] | 2 | 3000 |
| CR [Hu and Liu, 2004] | 2 | 376 |
| *Topic classification* | | |
| AGNews [Zhang et al., 2015] | 4 | 7600 |
| TREC [Voorhees and Tice, 2000] | 6 | 500 |
| *Natural language inference* | | |
| MNLI [Williams et al., 2018] | 3 | 3000 |
| RTE [Dagan et al., 2005] | 2 | 277 |
| *Reasoning* | | |
| ARC-Challenge [Clark et al., 2018] | 4 | 1170 |
| MMLU [Hendrycks et al., 2020] | 4 | 2000 |
| COPA [Roemmele et al., 2011] | 2 | 100 |
| *Word disambiguation* | | |
| WiC [Pilehvar and Camacho-Collados, 2019] | 2 | 638 |

## A.2  Implementation Details

**Experiments on internal mechanisms of biased factors**: All experiments are conducted on Llama-2 7b model. For the vanilla label bias experiment, we projecting all FFN value vectors into the vocabulary space and sum the label logits for all FFN vectors whose label logits rank within the top 10 over the vocabulary to calculate uncontextual accumulated FFN logits. We change different set of label words in prompt to derive the label prediction frequency of different label pairs. For the recency bias experiment, based on findings in [Wang et al., 2023a], instead of the summed attention weights over the whole example, we adopt the sum of attention weights on label words of the example, e.g. "Answer: positive" as the effective attention weight on each example. For the selection bias experiment, we use zeroshot ARC dataset prompts in Table 8, and we use 12 samples for each class. The attention weight is also summed on label words instead of the whole option.

**Baselines**:   We reproduce all baselines using the publicly available code released by the authors to ensure a fair comparison. For the PC method, instead of using test samples as in the original work, we employ 200 training samples per class as the estimate set for parameter estimation using the EM algorithm. This adjustment is made to reflect real-world scenarios where test samples are not readily available. Additionally, the number of samples used by the PC method is significantly larger than that used by our UniBias method.

**Unibias**:   In our method, all threshold values are determined through grid searching as described in the methodology section. Specifically, we use 20 samples per class as the support set for grid searching in all experiments. For each repetition of the experiment, the support set is randomly selected based on different random seeds. Additionally, to manipulate biased FFN vectors and attention heads, we create masks for the attention heads of all attention layers and adjust the FFN coefficient values and attention head masks using the hook operation. Additionally, we conduct the experiment on four A5000 GPUs.

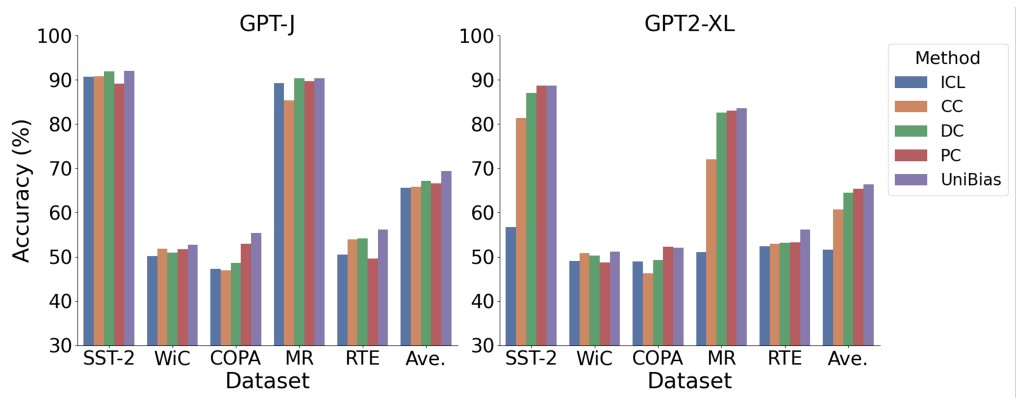

Figure 7: Performance comparison of our UniBias method against baseline methods using GPT-J and GPT2-XL models.

## B    Limitation and Future Work

In this work, we provide a novel insight into the internal mechanisms behind the bias of LLMs. As a pioneering effort in mitigating LLM bias through manipulation of the model's internal structures, our approach relies on grid searching with a small set of labeled training samples. Future research could focus on reducing this reliance, potentially improving the efficiency and applicability of our method.

There are many interesting avenues for future research. For instance, instead of identifying biased components for each ICL prompt, future work could explore the identification of global biased components that mitigate bias across multiple tasks and diverse prompt design settings. Additionally, the biased FFN vectors and attention heads we identify could potentially serve as sensors for guiding effective prompt generation. We expect that this internal perspective on LLM bias will inspire more innovative applications in both bias mitigation methods and prompt engineering.

## C    Evaluation on More LLMs

Figure 7 demonstrates the performance of various methods across multiple datasets when applied to GPT-J and GPT2-XL models. For both models, our UniBias method consistently outperforms the baseline methods including vanilla ICL, CC, DC and PC. Notably, the improvement on the GPT2-XL model is substantial, demonstrating over an over $20\%$ increase in accuracy on SST-2 dataset compared to vanilla ICL.

## D    Eliminating Common Biased Components

Table 5: List of common biased attention heads eliminated. Indexing Starts from 0.

| $(19, 10)$ | $(19, 14)$ | $(16, 29)$ | $(19, 21)$ | $(25, 21)$ | $(16, 11)$ | $(18, 31)$ | $(18, 1)$ |
|---|---|---|---|---|---|---|---|

We explore the potential of eliminating common biased components and apply it to diverse tasks, rather than addressing each task individually. We conduct additional experiments on multiple tasks to assess the effectiveness of directly elinimate these components. Experimental results in Table 2 indicate that although not as effective as our full Unibias method, it outperforms the vanilla ICL by a large margin. Notably, eliminating common biased components represents cost-free gain in performance, as it involves only the direct masking of biased components identified in our work and is applicable to diverse tasks. The attention heads that are masked are listed in Table 5.

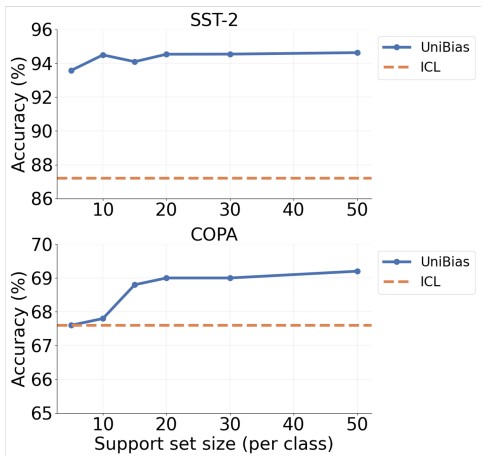

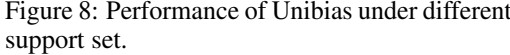

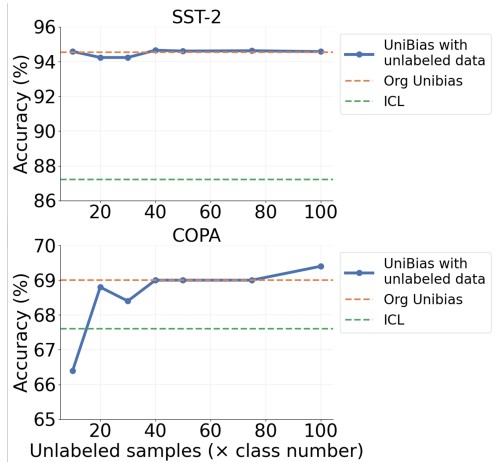

Figure 8: Performance of Unibias under different support set.

Figure 9: Performance of Unibias using unlabeled samples as support set. It is compared against standard ICL and the original Unibias.

## E Impact of Support Set Size

Our proposed UniBias method employs a small support set for grid searching. To analyze its effect, we vary the size of the support set. Figure 7 illustrates Unibias's performance with support set sizes ranging from 5 to 50 samples. The results indicate that the performance stabilizes when the support set contains 20 or more samples per class. Notably, for the SST2 dataset, even with much fewer support samples, Unibias significantly outperforms the standard ICL.

## F Using Unlabeled Samples for Support Set

To Address the potential challenge in accessing labeled samples, we further explore the alternative of using unlabeled samples during grid search. In our method, labeled samples are used to ensure each class is represented proportionally in the grid search, without direct use of the specific label information. Therefore, for balanced datasets, it is equally effective to employ a slight larger pool of unlabeled samples.

Our experimental findings, illustrated in Figure 9 of the rebuttal PDF, indicate that approximately $40 \times \#Classes$ unlabeled samples achieves performance comparable to that obtained with labeled samples.

## G Additional Analysis

We further analyze why mitigating model's bias towards labels can alleviate prompt brittleness in our method. Due to the inherent bias of LLMs, different prompts can lead to varying biases towards labels. For example, due to recency bias, placing a negative sentiment analysis sample at the end of a prompt can make LLMs tend to predict 'negative', incorrectly classifying positive samples and thus degrading ICL performance. Various bias factors lead to different direction and extend of bias, resulting in different changes in ICL performance and leading to the prompt brittleness. In contrast, our UniBias method effectively mitigates various potential biases inherent in LLMs by addressing their root causes internally from LLMs. By doing so, it minimizes the introduction of bias towards labels regardless of the difference in prompts, leading to more stable and accurate ICL performance across different prompt configurations.

Additionally, our UniBias method seeks to address a broad range of factors that lead to LLM bias, extending beyond those discussed in Section 2. Given the significant variability in prompts, models, and data corpuses, numerous unanticipated bias factors may emerge. Our approach is designed to

tackle these diverse bias factors comprehensively. This is feasible because biased behaviors observed externally in LLMs originate from their internal components—specifically, the feedforward neural network (FFN) vectors and attention heads, which house nearly all LLM parameters. By directly identifying and mitigating biases within these FFN vectors and attention heads, UniBias offers a foundational strategy to counteract various forms of bias.

# H   Prompt Templates

The prompt templates used in this work are provided below. We generate few-shot ICL templates follow the template styles in [Han et al., 2023, Fei et al., 2023], as illustrated in Table 6.

Table 6: Prompt templates for all $k$-shot ICL experiments.

| Dataset | Template | Label Space |
|---|---|---|
| SST-2 CR MR | Review: {*sentence*} 
 Sentiment: {*label*} | negative / positive |
| MNLI | Premise: {*premise*} 
 Hypothesis: {*hypothesis*} 
 Answer: {*label*} | yes / maybe / no |
| ARC MMLU | Question: {*question*} 
 {*options*} 
 Answer: {*label*} | A / B / C / D |
| SST-5 | Review: {*sentence*} 
 Sentiment: {*label*} | terrible / bad / okay / good / great |
| AGNews | Article: {*passage*} 
 Answer: {*label*} | world / sports / business / technology & science |
| TREC | Question: {*sentence*} 
 Answer Type: {*label*} | abbreviation / entity / description / person 
 / location / number |
| COPA | Premise: {*premise*} 
 Choice1: {*choice1*} 
 Choice2: {*choice2*} 
 Answer: {*label*} | 1 / 2 |
| RTE | Premise: {*sentence1*} 
 Hypothesis: {*sentence2*} 
 Answer: {*label*} | yes / no |
| WiC | Sentence1: {*sentence1*} 
 Sentence2: {*sentence2*} 
 Word: {*word*} 
 Answer: {*label*} | false / true |

Table 7: Templates of different prompt formatting used in the prompt brittleness experiment for SST-2.

| ID | Template | Label Space |
|---|---|---|
| 1 | Review: {Sentence}
Sentiment: {Label} | Positive / Negative |
| 2 | Input: {Sentence}
Prediction: {Label} | Positive / Negative |
| 3 | Review: {Sentence}
Sentiment: {Label} | good / bad |
| 4 | {Sentence} It was {Label} | good / bad |
| 5 | Review: {Sentence}
Positive Review: {Label} | Yes / No |
| 6 | {Sentence} My overall feeling was that the movie was {Label} | good / bad |
| 7 | Review: {Sentence}
Question: Is the sentiment of the above review Positive or Negative?
Answer: {Label} | Positive / Negative |
| 8 | My review for last night's film: {Sentence}The critics agreed that this movie was {Label} | good / bad |

Table 8: Prompt templates for the 0-shot experiments.

| Dataset | Template | Label Set |
|---|---|---|
| SST-2 | Review: {sentence}
Sentiment: {label} | negative / positive |
| COPA | Premise: {premise}
Choice1: {choice1}
Choice2: {choice2}
Answer: {label} | 1 / 2 |
| MMLU | Question: {*question*}
{*options*}
Answer: {*label*} | A / B / C / D |

