# OpenReview forum: "UniBias: Unveiling and Mitigating LLM Bias through Internal Attention and FFN Manipulation"
_NeurIPS.cc/2024/Conference — NeurIPS 2024 poster_

### Official Review · Reviewer_CnNd · 2024-07-02

**Soundness:** 3
**Presentation:** 2
**Contribution:** 3
**Rating:** 6
**Confidence:** 4

**Summary:**

This paper addresses internal biases in LLMs that cause prompt brittleness. Unlike previous external adjustment methods, it examines the roles of MLPs and attention heads in creating these biases. The proposed UniBias method identifies and masks these biased components, thereby enhancing the model's in-context learning ability. Experiments across several NLP tasks demonstrate that UniBias effectively improves ICL performance.

**Strengths:**

* This paper explores the internal contributions of FFN vectors and attention heads to LLM bias, a relatively unexplored area.
* The proposed method is straightforward and does not introduce any additional inference time cost, unlike post-calibration methods.

**Weaknesses:**

The main weakness of this paper is that there seems to be some disconnect between the problem it aims to solve (the prompt sensitivity of large language models, or LLMs) and two of the objectives of the preliminary analysis (how internal components generate common token bias and selection bias towards labels).

Specifically, the challenge the paper proposes to address is the sensitivity of LLMs to input prompts (example selection, format, and order), and this issue is also illustrated in Figure 1. However, two of the targets of the exploration section are the biases present when LLMs use labels from inputs.

The authors do not explicitly state why suppressing the model's bias toward labels can simultaneously alleviate the model's bias toward prompts. For example, intuitively, suppressing the label's preference for option A does not necessarily suggest an improvement in the model's bias towards a certain format or example.

Given this lack of clarity, and since the criteria for detecting biased components also focus on labels, it is unclear why the proposed method can reduce prompt brittleness.

**Questions:**

NA

**Limitations:**

Limitations: Yes; Broader impacts: NA

---

> ### Author Rebuttal · Authors · 2024-08-07
>
> Dear Reviewer CnNd,
>
> We deeply appreciate your thorough review and valuable feedback. Below is a summary of our answers (**A**) to the weaknesses (**W**) you raised.
>
> ---
> **[W1]**:  There seems to be some disconnect between the problem it aims to solve and two of the objectives of the preliminary analysis.
>
> **[A1]**: Thank you for raising this important issue! We apologize for the lack of clarity in linking these aspects in our paper.
>
> The connection between prompt sensitivity of LLMs and their bias towards labels is well-established in the LLM bias mitigation literature and we mentioned this in Line 21 of our paper. However, thanks to your insightful feedback, we realize the need to more thoroughly strengthen and elucidate this connection to enhance understanding of our method's effectiveness. A detailed analysis is provided below:
>
> **The Connection Established by Literature**: Firstly, the literature demonstrates that the prompt sensitivity arises from LLMs' inherent bias towards predicting certain answers. Therefore, the prompt brittleness can be addressed by mitigating LLMs' bias towards labels. For example, in the classic study of "Calibrate Before Use" [1] it explicitly states that:
> > We demonstrate that this instability of prompt arises from the bias of language models towards predicting certain answers.
>
> They detailed this analysis in Section 4 of their paper. Building on this foundation, following studies address the bias of LLMs toward to address prompt brittleness. For example, a domain-label bias is identified and mitigated to alleviate prompt brittleness in [2].
>
> **Additional Analysis on Our Method**: We further analyze why mitigating model's bias towards labels can alleviate prompt brittleness in our method. Due to the inherent bias of LLMs, different prompts can lead to varying biases towards labels. For example, due to recency bias, placing a negative sentiment analysis sample at the end of a prompt can make LLMs tend to predict 'negative', incorrectly classifying positive samples and thus degrading ICL performance. Various bias factors lead to different direction and extend of bias, resulting in different changes in ICL performance and leading to the prompt brittleness. In contrast, our UniBias method effectively mitigates various potential biases inherent in LLMs. By doing so, it minimizes the introduction of bias towards labels regardless of the difference in prompts, leading to more stable and accurate ICL performance across different prompt configurations.
>
> **Experimental Validations**: We further provide experimental results to demonstrate how mitigating bias towards labels can alleviate prompt brittleness in ICL. We analyze the average prediction probabilities of 'negative' and 'positive' across all test samples in various prompt formats on the SST2 dataset. The ICL performance corresponding to these formats is depicted in Figure 1 of our paper. As shown in the table below, the average prediction probabilities for each format are represented as $(P_{\text{ave}}(\text{neg}), P_{\text{ave}}(\text{pos}))$. Our results reveal that different prompt formats introduce varying biases, leading to different levels of label bias which, in turn, can destabilize ICL performance. Conversely, our UniBias method produces balanced probabilities across labels and significantly mitigates label bias, resulting in more stable and accurate ICL performance and reducing prompt brittleness.
>
> |Method|Format 1|Format 2| Format 3|Format 4|Format 5|
> |--|--|--|--|--|--|
> | Vanilla ICL| (0.34, 0.55) | (0.35, 0.61) | (0.31, 0.40) | (0.33, 0.39)|(0.33, 0.55)|
> | UniBias | (0.49, 0.49) | (0.47, 0.48) | (0.35, 0.35) |(0.40, 0.39) |(0.46,0.48)|
>
> We appreciate your constructive comments, which have significantly enhanced the clarity of our paper. We will make sure to incorporate the above discussions into the revised manuscript.
>
> ---
> Moreover, we would like to highlight other enhancements made during the rebuttal process:
> * **Expanded LLM Evaluation**: We conduct additional experiments to evaluate UniBias on more LLMs, including *GPT-J (6B)* and *GPT2-XL (1.5B)*. Detailed results can be found in *Table 1* of the attached one-page PDF.
> * **Exploration of Common Biased Components Across Tasks**: We identify and eliminate the common biased components within LLMs and evaluate its performance on multiple tasks. Results are detailed in *Table 4* of the new PDF. This experiment demonstrates the potential of our work in stimulating diverse bias mitigation methods in a novel perspective of LLM inner structure manipulation.
> * **Enhancement in Grid Search Process**: We optimize the grid search process used in our method by providing alternatives of implementing one-time grid search and using unlabeled samples in place of labeled ones. The results of these optimizations are depicted in *Table 2* and *Figure 1* of the rebuttal PDF.
> * **In-depth Component Analysis**: We further visualize and analyze the distribution of identified biased attention heads, as shown in *Figure 2* in the PDF.
>
> ---
> We are encouraged to find that these additioanl analysis and experiments, prompted by your valuable insights, have substantially strengthened our paper. We look forward to hearing your feedback!
>
> ---
> **References**:
>
> [1] Calibrate before use: Improving few-shot performance of language models. ICML 2021.
>
> [2] Mitigating Label Biases for In-context Learning. ACL 2023.

---

> > ### Comment · Reviewer_CnNd · 2024-08-10
> >
> > Thank you for your comprehensive response. I believe these discussions and explanations indeed serve as excellent supplements to clarity. However, before I finalize my decision, I feel there is still one point that hasn't been covered by these discussions. Even after we've determined the label names, examples, and their order, there's still significant room for variation in the other parts of the prompt beyond these elements. For instance, the way queries and labels are connected, the method of separating examples, the overall format of the input, and so on. These variations, beyond what has already been discussed, can also lead to substantial fluctuations in model performance [1]. However, this situation cannot be explained by any of the three types of bias discussed in this work.
> >
> > [1] Sclar, Melanie, et al. "Quantifying Language Models' Sensitivity to Spurious Features in Prompt Design or: How I learned to start worrying about prompt formatting." The Twelfth International Conference on Learning Representations.

---

> ### Author Response · Authors · 2024-08-10
> **Response to Reviewer CnNd**
>
> Dear Reviewer CnNd,
>
> Thank you for your prompt response and for recognizing the discussions presented in our rebuttal! We greatly appreciate your insightful feedback regarding the broad spectrum of prompt variations that can lead to fluctuations in model performance. We are glad to further discuss this important issue.
>
> We completely agree with your insight about the significant room for variation across different parts of a prompt. Actually, this insight has been a key motivation for the development of our UniBias method. We believe that merely identifying and addressing a limited number of specific bias factors based on superficial observations is not optimal or entirely effective for mitigating LLM bias. Indeed, due to the significant room in variations in prompts/models/data corpus, it is nearly impossible to identify and address all bias factors from just analyzing external effects of prompt variations on model performance. Therefore, the key challenge and limitation of current bias mitigation methods lie in designing a method that can universally address all potential factors contributing to LLM bias and fluctuations ICL performance.
>
> This challenge motivates the rationale for our UniBias method. UniBias is designed to address all potential biases, whether arising from prompt variations, data corpus discrepancies, or model variations. This is feasible because, **fundamentally, all these superficial variations lead to biased behaviours internally in either the FFN vectors or the attention heads**—components where nearly all LLM parameters reside. By directly identifying and mitigating biases within these FFN vectors and attention heads, UniBias provides a foundational approach to address all forms of bias, representing a poineering work in achieve this objective and a significant advancement in LLM bias mitigation.
>
> The three types of bias factors discussed in the paper are intended to demonstrate the internal mechanisms of these biases observed superficially. Given the vast potential for variations in prompt-induced biases, as you noted, it would be impractical to discuss the internal mechanisms for every possible bias factor.
>
> Thank you for highlighting this crucial point, which not only validates a very important strength of our method but also aligns perfectly with our motivation for developing it. We intend to include a discussion on this issue in our revised manuscript.

---

> > ### Comment · Reviewer_CnNd · 2024-08-11
> >
> > Thank you for your further clarification. I agree that addressing as many types of bias as possible is important, and consequently, debiasing the models internally might be the most fundamental approach. I also agree that it's impossible to analyze every type of bias. However, I believe there may be some misunderstanding about my concerns.
> >
> > My remaining concern is specifically about the prompt format bias that I mentioned in my previous comment, which neither your provided literature nor your additional response has addressed. I believe this bias is quite significant but not covered by the three biases discussed in your paper, as mainly evaluated in [1]. By saying this bias is significant, I mean that models are also highly sensitive to prompt formatting, not that there is too much room for prompt formatting to discuss in detail.
> >
> > Therefore, when the claim states "effectively mitigating the models' prompt sensitivity," I assume this type of bias is also addressed. I think the following questions might help you understand my concerns and the discussion I expect to see in the paper more clearly:
> >
> > 1. Does the prompt sensitivity issue that the paper claims to mitigate cover the prompt format bias?
> >
> > 2. If so, since this bias is not discussed, what is your speculation as to why the proposed method can mitigate this? Or are you unsure whether the method indeed mitigates this type of bias since you haven't analyzed it in the paper, and thus the method is not explicitly addressing this?
> >
> > 3. If not, I think it's necessary to discuss more about the scope of prompt sensitivity, explicitly stating what is not addressed by the proposed method.

---

> > > ### Author Response · Authors · 2024-08-11
> > > **Response to Reviewer CnNd**
> > >
> > > Dear Reviewer CnNd,
> > >
> > > Thank you for your follow-up and for clarifying your concerns regarding prompt format bias. We are pleased to the common ground we share and appologize for any misunderstanding. Your insights are invaluable in refining the depth and clarity of our work!
> > >
> > > We appreciate the questions you provided, as they are very helpful in facilitating our discussion. We would like to respond by addressing these questions.
> > >
> > > ---
> > > Firstly, our method can mitigate prompt format bias, which is supported by experimental evidence in our paper. We evaluate the ICL performance of both the vanilla LLM and the LLM employing our UniBias method under various prompt formats. These experimental results are depicted in the middle figure of Figure 1 in the paper, with different prompt templates used in this experiment detailed in Table 5 of the Appendix. The prompt formats we evaluated include many of the formats discussed in [1], such as differences in separators (': ' vs ' '), spacing ('\n' vs ' '), casing (Positive/positive—with 'positive' used in prompts in Table 4 of the Appendix), label names (Positive/good/Yes), question formats, and the overall format of the prompts.
> > >
> > > From Figure 1 of the paper, we can observe that these variations in prompt formats indeed lead to significant fluctuations in ICL performance of the vanilla LLM, as suggest by you and [1]. In contrast, despite these challenging prompt format variations, our UniBias method consistently achieve much more accurate and stable ICL performance across all tested formats.
> > >
> > > ---
> > > Next, we address the important question of why UniBias effectively mitigate prompt format bias:
> > >
> > > - As concluded in [2] and analyzed our rebuttal, prompt instability arises from the LLMs' bias towards predicting certain answers. Different prompt formats can induce biases of varying directions and extents, leading to different effects of bias, and consequently, fluctuations in ICL performance. This is also evidenced by the table provided in the rebuttal, which shows average prediction probabilities for the five prompt formats evaluated in the prompt formatting experiment in Figure 1 of the paper.
> > > - On the other hand, recalling our analysis on mechanistic interpretability of LLMs (detailed in Section 2.1 of the paper): each attention layer  FFN layer contribute to the final prediction by adding their output hidden states to the residual stream. These outputs can be viewed as the sum of their respective attention heads and FFN vectors.
> > >
> > > Integrating these two insights, we can learn that the bias of the overall LLM introduced by prompt format bias can be attributed to biased behaviors internally in either the FFN vectors or the attention heads—components where nearly all LLM parameters reside. Therefore, by directly identifying and mitigating biases within these FFN vectors and attention heads, UniBias provides a foundational approach to address prompt format bias and its resultant fluctuations in ICL performance. Because of the same rationale, UniBias is not only effective against the three types of bias discussed in our paper but is also capable of mitigating prompt format bias, example selection bias (as analyzed in Figure 1), and other potential bias factors. This reflects our objective, as discussed in this discussion thread, of designing a method capable of universally addressing all bias factors that contribute to LLM bias.
> > >
> > > Finally, we want to emphasize that the your insightful question is crutial in demonstrating why Unibias works. We intend to include a detailed subsection in of our revised manusprict to thoroghly discuss this issue.
> > >
> > > Thank you again for your thoughtful engagement, which is instrumental in refining our presentation and deepening our analysis. We truly appreciate these inspiring discussions.
> > >
> > > ---
> > > [1] "Quantifying Language Models' Sensitivity to Spurious Features in Prompt Design or: How I learned to start worrying about prompt formatting." ICLR 2024.
> > >
> > > [2] Calibrate before use: Improving few-shot performance of language models. ICML 2021.

---

### Official Review · Reviewer_8E5c · 2024-07-11

**Soundness:** 4
**Presentation:** 3
**Contribution:** 3
**Rating:** 6
**Confidence:** 4

**Summary:**

This paper aims to address prompt brittleness in in-context learning methods introduced by seemingly inconsequential changes such as vanilla label bias, recency bias, and selection bias. They use the logit lens technique, where  the linear function is applied from the final layer of a decoder-only transformer (AKA the unembedding matrix) on the intermediate layers to interpret the attention activations and FFN embeddings. They detect biased activations and FFN vectors by their relatedness, residual bias, and low variance. Applying their method results in unchanged or improved performance compared to the baseline.

**Strengths:**

- The paper introduces a novel application of mechanistic interpretability to mitigate biases in in-context learning and demonstrates that specific attention heads and FFNs in the decoder can be isolated as responsible for prompt brittleness.

- The methodology and findings are insightful and bring consistent performance improvements.

- The experiments are extensive, covering various datasets and settings, providing strong evidence for the effectiveness of the proposed methods to support the conclusions.

- The paper is in general well-written, with clear explanations of the methodology and results.

**Weaknesses:**

- The paper mentions some tasks could share biased components (line 318-321) – based on their experiments, I think it would have been possible to identify these components in this work, and I would have liked to see this result.

- The analysis of their results could be expanded further. In table 2 the performance of FFN-only, attention-only, and their full UniBias method is inconsistent – sometimes FFN-only performs the best, sometimes attention-only, with no pattern on dataset task.

- It would be interesting to see how bias exists across different layers of the model. Analyzing the distribution and impact of biased components in each layer may provide deeper insights into how biases are propagated and amplified throughout the model.

- Some of the mathematical formulations, particularly those related to the identification of biased components, could be more detailed to enhance reproducibility and understanding. The authors could provide more thorough mathematical derivations and examples illustrating how the bias criteria are identified. This would make the methodology clearer and easier to understand.

**Questions:**

- Do you have any insights on why UniBias improves performance on certain tasks but not others?

- For vanilla label bias, how do you choose the alternative label names?

- It is interesting that the unembedding matrix can also be applied to the attention hidden state. Did you notice any discrepancies in the logit distribution between the FFN and the attention head of the same layer?

- The sections and overall flow of Sections 2 and 3 may be organized more clearly

  - Sections 3.3 and 3.4 seem very small (3 and 11 lines respectively) and could be combined with other sections.

  - It is unclear how the criterion they introduce (relatedness, bias, low variance) in lines 195-201 relate to the three bias conditions introduced in line 218

  - In general, Sections 2 and 3 seem related to each other but the writing does not tie them together clearly (eg. lines 202-205 seem to be describing the importance of low variance to identifying the biases in section 2.2, but it is not stated very clearly)

- There are further minor writing issues that could be revised, including various minor typos.

**Limitations:**

Yes.

---

> ### Author Rebuttal · Authors · 2024-08-07
>
> Dear Reviewer 8E5c,
>
> We deeply appreciate your thorough review and valuable feedback. Below is a summary of our answers (**A**) to the weaknesses (**W**) and questions (**Q**) you raised.
>
> ---
> **[W1]**: Show shared biased components across tasks.
>
> **[A1]**: We greatly value your suggestion! Inspired by your insights, we not only list the common biased components, but also explore eliminating common biased components for bias mitigation.
>
> **Listing of Common Biased Components**:  The shared biased attention heads with their frequency of occurrence (>=3) are detailed in *Table 3* of the rebuttal PDF. The shared biased FFN vectors are not discussed because they are associated with label words, which typically vary across tasks.
>
> **Eliminating Common Biased Components**: Inspired by your feedback, we explore the potential of eliminating common biased components and apply it to all tasks, rather than handling each task individually. We conduct additional experiments on multiple tasks to assess the effectiveness of directly elinimate these components. Experimental results in *Table 4* of the new PDF indicate that although not as effective as our full Unibias method, it outperforms the vanilla ICL by a large margin. Notably, this is a free gain in performance as it involves merely the direct masking of the biased components identified in our work and is applicable to diverse tasks.
>
> ---
> **[W2]**: Result analysis could be expanded for Table 2.
>
> **[A2]**: We deeply appreciate your suggestion and provide a detailed analysis here.
>
> As an ablation experiment, we find that both FFN-only and attention-only methods outperform the standard ICL, demonstrating their effectiveness. However, as you suggested, there is no consistent pattern on when these two ablation methods perform better. The removal of biased FFN vectors predominantly addresses uncontextual biases associated with label words, while the removal of biased attention heads targets contextual biases, such as recency bias. Therefore, the relative effectiveness of these methods depends on the dominance of either type of bias in the given task and the specifics of the prompt examples, which can vary unpredictably.
>
> Despite these variations, it is notable that our comprehensive UniBias method consistently yields the best performance, underscoring the effectiveness and robustness of UniBias.
>
> ---
> **[W3]**: It would be interesting to see how bias exists across different layers of the model.
>
> **[A3]**: Thanks for your insightful suggestion! We visualize the identified biased attention heads in *Figure 2* of the new PDF and provide analysis below. Biased FFN vectors are not visualized due to the high dimensionality of FFN layers.
>
> In *Figure 2*, the intensity of color corresponds to the frequency with which an attention head is identified as biased across 12 datasets, each repeated 5 times. The visualization reveals that biased attention heads mostly occur in the middle layers. This observation aligns with the established understanding of LLM architecture: early layers tend to encode shallow and discrete concepts  such as lexical features, while higher layers are responsible for high-level semantics and complex reasoning. Therefore, discrete concepts at early layers are less likely to directly contribute to the bias. In contrast, middle layers, where simple reasoning begins to form are more likely to introduce biases (e.g., "A sentence is always positive rgardless of its content"). The higher layers are less likely to produce this kind of bias associated with simpler, shallow reasoning.
>
> ---
> **[W4]**: Some mathematical formulations can be more detailed.
>
> **[A4]**: We appreciate the valuable suggestion! The formulations for identifying biased components involve measuring the sum of label logits, the bias of label logits, and the coefficient variance of FFN vector coefficients or attention head label logits. They are corresponding to the relatedness, biased, and low variance criterions. Based on your feedback, we will include more comprehensive mathematical derivations and examples in the revised paper to clearly illustrate how these bias criteria are identified and applied.
>
> ---
> *We apologize for the brevity of the following responses due to space limitations.*
>
> ---
>
> **[Q1]**: Unibias performance
>
> **[A5]**: Thank you for this important question. To address it, it's essential to consider that a more appropriate comparison for UniBias would be between standard ICL and UniBias, as this comparison contrasts ICL based on vanilla LLMs with LLMs after the elimination of biased components. In this case, our UniBias consistently improve performance across all 12 datasets we evaluated. On the other hand, calibration methods are under different methodology and perform better on a limited number of datasets. This may be attributed to the specific characteristics of each dataset.
>
> ---
> **[Q2]**: Alternative label names for vanilla label bias
>
> **[A6]**: They are generated by GPT-4.
>
> ---
> **[Q3]**: Discrepancies in logit distribution between FFN and attention head
>
> **[A7]**: Thank you for the insightful question. Generally, they are consistent across the middle and higher layers. However, we observe  discrepancies in some individual components.
>
> ---
> **[Q4]**: Organization of Sections 2 and 3.
>
> **[A8]**: Thank you for the detailed suggestions! We will revise Sections 2 and 3 carefully. This includes merging Sections 3.3 and 3.4, enhancing coherence between the criteria and conditions, and strengthening connections between 3.3 and 3.4. We will also ensure a thorough review for typos and writing issues as suggested in **Q5**.
>
> ---
> We wish to highlight that we also **extend the evaluation of UniBias** to include *GPT-J* and *GPT2-XL*, which are detailed in *Table 1* of the new PDF.
>
> ---
> We are encouraged to find that these experiments and enhancements, prompted by your valuable insights, have substantially strengthened our paper. We look forward to hearing your feedback!

---

### Official Review · Reviewer_LH5i · 2024-07-14

**Soundness:** 1
**Presentation:** 2
**Contribution:** 2
**Rating:** 5
**Confidence:** 4

**Summary:**

This paper seek to address prediction bias in ICL through intervening feedforward vectors and attention heads.

**Strengths:**

* This paper studies a critical problem of in ICL.
* The proposed method is well-motivated.

**Weaknesses:**

The major problem with the proposed method is that it requires approximately 20 labeled instances per class. This leads to the following concerns:
* The comparison between some baselines and the proposed method is unfair. The proposed method requires labeled instances for searching thresholds, while baselines like DC require no labeled instances.
* The generalizability of the proposed method is also questionable. Can we search thresholds once and apply them to all datasets? If not, it is impractical for real-world application. If we have that many labeled instances and access to the model parameters for the target task and dataset, we can always use them to finetune the model to achieve better performance.
* Moreover, the cost will be intractable when the number of classes increase.

**Questions:**

See weaknesses.

**Limitations:**

This paper does not have a limitation section.

---

> ### Author Rebuttal · Authors · 2024-08-07
>
> Dear reviewer LH5i,
>
> We deeply appreciate your thorough review and valuable feedback. We appreciate your acknowledgment of the significance of the problem we investigated and motivation behind our approach. Below is a summary of our answers (**A**) to the weaknesses (**W**) you raised.
>
> ---
> **[W1]**: Grid search with 20 labeled samples per class can lead to concerns in fairness in comparison with baselines, generalizability of the method, and increased cost when the number of classes is large.
>
> **[A1]**: Thank you for raising this important concern! It prompted us to further enhance our method.
>
> Initially, we choose a limited number of labeled samples for grid search because it is typically manageable given the small size of labeled data. Additionally, it is only needed for the initial set up to identify biased LLM components. During inference stage, the computation cost of UniBias is completely identical to that of the vanilla LLMs. We believe this is a reasonable trade-off for the substantial improvements in ICL performance and the novel perspective Unibias offers on mitigating bias through LLM inner components.
>
> However, we understand the concern that using labeled samples can be challenging when labeled data are scarce or the number of classes is large. Therefore, we have optimized our method in light of your valuable feedback:
>
> **Replacing Labeled Samples with Unlabeled Ones**: To Address the potential challenge in accessing labeled samples and ensure a fair comparison with baselines, we further explore the alternative of using unlabeled samples during grid search. In our method, labeled samples are used to ensure each class is represented proportionally in the grid search, without direct use of the specific label information. Therefore, for balanced datasets, it is equally effective to employ a slight larger pool of unlabeled samples.
>
> Our experimental findings, illustrated in *Figure 1* of the rebuttal PDF, indicate that approximately $40 \times \text{number of classes}$ unlabeled samples achieves performance comparable to that obtained with labeled samples. This number is significantly lower than the quantity required by the PC baseline, which requires at least  $200 \times \text{number of classes}$ unlabeled samples.
>
> **Optimization of Grid Search**: Inspired by your feedback, we streamline the grid search process. We now provide an alternative to perform grid search on a single dataset to obtain a set of threshold values, which are then applied universally across other datasets. This alternative eliminates the need for repeated grid searches across each dataset, significantly enhancing efficiency and scalability of the method. By directly adopting the fixed threshold, it also addesses the scalability issues associated with an increasing number of classes.
>
> Our experimental results in *Table 2* of the rebuttal PDF confirm that using a fixed set of thresholds maintains good performance, slightly below the original UniBias method but still superior to vanilla ICL and other baseline methods. This also illustrate the robustness of our method in terms of threshold selection.
>
> **Exploration of Eliminating Common Biased Components**: In response to your feedback, we have delved deeper into the potential of eliminating common biased components and apply it to all tasks, rather than handling each task individually. In Section 4.4 of our paper, we discuss the presence of shared biased LLM components across different tasks. Inspired by this finding, we conduct additional experiments to directly elinimate these common biased components and evaluate on different tasks.
>
> Experimental results in *Table 4* of the rebuttal PDF indicate that although not as effective as our full Unibias method, it outperforms the vanilla ICL by a large margin. Notably, this is a free gain in performance as it involves merely applying a direct masking to the biased components identified in our work and is applicable to diverse tasks.
>
> Finally, following your suggestion, we evaluate the performance of fine-tuning LLMs using 20 labeled samples per class and find that their performance is significantly lower than that of UniBias. This results further validates the effectiveness of our method.
>
> | |SST2 |MNLI|Trec
> |--|--|--|--|
> | ICL | 87.22 | 53.83 |72.92|
> | ICL after SFT |88.64 | 53.95  |73.2|
> | UniBias | **94.54** |**54.97**|**80.80**|
>
> The enhancements suggested by your feedback have markedly improved the efficiency and scalability of our approach. Furthermore, the third experiment underscores the potential of our work in stimulating diverse bias mitigation methods in a novel perspective of LLM inner structure manipulation.
>
> ---
> During rebuttal, we also **extend the evaluation of UniBias** to include *GPT-J* and *GPT2-XL*, which are detailed in *Table 1* of the new PDF.
>
> ---
> We would like to further highlight the contributions of our work.
> * **A New Insight for LLM Bias Mitigation**: Unlike existing works based on *external* adjustments of LLM outputs, we mitigate LLM bias through manipulation of LLM *internal* structure. This novel perspective could potentially stimulate future research on LLM bias mitigation from inner structure manipulation, offering a new direction for the field.
> * **Unveil Internal Mechanisms of LLM Bias**: We conduct a thorough exploration of the internal mechanisms underlying biases in LLMs, providing deep insights that go beyond superficial observations, revealing the inner causes of these biases.
> * **Extensive Evaluation**: UniBias is evaluate across 12 datasets and 4 LLMs, consistently demonstrating its effectiveness.
> * **Addressing Prompt Brittleness**: Our experimental results show that our method effectively mitigates prompt brittleness, a critical issue in ICL.
>
> ---
> We are encouraged to find that these experiments and enhancements, prompted by your valuable insights, have substantially strengthened our paper. We look forward to hearing your feedback!

---

### Official Review · Reviewer_BhZE · 2024-07-16

**Soundness:** 3
**Presentation:** 3
**Contribution:** 3
**Rating:** 6
**Confidence:** 3

**Summary:**

The paper explores the influence of FFNs and attention heads in LLMs on biases, resulting in model predictions that exhibit favoritism towards specific labels. The authors propose UniBias, an inference-only technique designed to detect and mitigate biased components within LLMs by analyzing and manipulating FFN vectors and attention heads.  Experiment results on 12 diverse NLP datasets show that UniBias significantly improves the in-context learning performance of LLMs and reduces their sensitivity to prompt design.

**Strengths:**

- The paper introduces UniBias, a novel method designed to mitigate bias in LLMs by manipulating internal model components, specifically FFN vectors and attention heads. This approach represents a unique contribution to bias reduction within LLMs.
- UniBias demonstrates significant improvements over standard ICL and state-of-the-art calibration methods, indicating its potential to become a leading technique in bias mitigation for LLMs.
- The authors conduct an in-depth exploration of the internal mechanisms underlying biases in LLMs, providing insights that delve beyond superficial observations to uncover the root causes of these biases.

**Weaknesses:**

- While the method shows promising results, it is crucial to assess how well these findings generalize across different types of LLMs and datasets (e.g., machine translation tasks) beyond those tested. This paper only conducts experiments on Llama and classification tasks.
- The reliance on grid search with a limited number of labeled training samples for identifying biased components may limit scalability and efficiency.

**Questions:**

- Is this analytical method compatible with LLMs that have undergone SFT? If so, how would the fine-tuning process potentially influence the identification and mitigation of biases within the model's internal mechanisms?
- Can the grid search process be automated or optimized to minimize the need for manual parameter tuning?

---

> ### Author Rebuttal · Authors · 2024-08-07
>
> Dear Reviewer BhZE,
>
> We deeply appreciate your thorough review and valuable feedback. Below is a summary of our answers (**A**) to the weaknesses (**W**) and questions (**Q**) you raised in the review.
>
> ---
> **[W1]**: Assess how well the findings generalize across different types of LLMs and datasets.
>
> **[A1]**: We greatly value your insightful suggestion and we have conducted additional experiments to validate our method.
>
> **Model Generalization**: Following your suggestion, we have conducted additional experiments to evaluate more LLMs, including *GPT-J (6B)* and *GPT2-XL (1.5B)*. The experimental results are detailed in *Table 1* of the rebuttal PDF. Together with the *Llama2-7B* and *Llama2-13B* models evaluated in our paper, we rigorously evaluate our method across four popular LLMs of different sizes and demonstrated the effective of our method across these models.
>
> **Task Coverage**: Our work is currently focus on classification tasks, consistent with the prevailing LLM bias mitigation literature. Our work evaluates extensive array of tasks compared to the exsiting studies. To provide context, current works typically focus either on mitigating bias in general classification tasks [1-2] or exclusively address bias in multiple-choice questions (MCQs) in reasoning tasks [3-4]. In contrast, our study covers 12 datasets spanning 5 distinct tasks, including both general classification tasks and MCQs in reasoning tasks. This breadth not only aligns with but surpasses the scope of tasks examined in the literature. Moving forward, we are enthusiastic about adapting our method to a wider variety of tasks as you suggested.
>
> We appreciate your insights, which significantly strengthened our experimental validation.
>
> ---
> **[W2]**: Grid search with a limited number of labeled samples may limit scalability and efficiency
>
> **[A2]**: Thank you for raising this concern, which prompted us to further enhance our method. Although using a small set of labeled samples is typically manageable, we recognize the potential for improvement. In response, we conduct experiments to address your concern from three aspects.
>
> **Optimization of Grid Search**: To streamline the grid search process, we now provide an alternative to perform grid search on a single dataset to obtain a set of threshold values, which are then applied universally across other datasets. This alternative eliminates the need for repeated grid searches across each dataset, significantly enhancing efficiency and scalability of the method.
>
> Our experimental results in *Table 2* of the rebuttal PDF confirm that using a fixed set of thresholds maintains good performance, slightly below the original UniBias method but still superior to vanilla ICL and other baseline methods. This experiment also illustrate the robustness of our method in terms of threshold selection.
>
> **Replacing Labeled Samples with Unlabeled Ones**: To Address the potential challenge in accessing labeled samples, we further explore the alternative of using unlabeled samples during grid search. In our method, labeled samples are used to ensure each class is represented proportionally in the grid search, without direct use of the specific label information. Therefore, for balanced datasets, it is equally effective to employ a larger pool of unlabeled samples.
>
> Our experimental findings, illustrated in *Figure 1* of the rebuttal PDF, indicate that approximately $40 \times \text{number of classes}$ unlabeled samples can achieve performance comparable to that obtained with labeled samples.
>
> **Exploration of Eliminating Common Biased Components**: In response to your feedback, we have delved deeper into the potential of eliminating common biased components and apply it to all tasks, rather than handling each task individually. In Section 4.4 of our paper, we discuss the presence of shared biased LLM components across different tasks. Inspired by this finding, we conduct additional experiments to directly elinimate these common biased components and evaluate on different tasks.
>
> Experimental results in *Table 3* of the rebuttal PDF indicate that although not as effective as our full Unibias method, it outperforms the vanilla ICL by a large margin. Notably, this is a free gain in performance as it involves merely applying a direct masking to the biased components identified in our work and is applicable to diverse tasks.
>
> The enhancements suggested by your feedback have markedly improved the efficiency and scalability of our approach. Furthermore, the third experiment indicates the potential of our work in stimulating future bias mitigation methods in a novel perspective of LLM inner structure manipulation.
>
> ---
> **[Q1]**: Compatibility with SFT
>
> **[A3]**: Thank you for the interesting question. We would like to respond in two aspects: the performance of our method on fully fine-tuned models and the impact of SFT on biased components.
>
> **Compatibility with SFT**: UniBias is fully compatible with LLMs that have undergone SFT. We applied UniBias to Llama2-7b fine-tuned on the SST2 and MNLI datasets, with following experimental results:
>
> ||SST2|MNLI
> |--|--|--|
> |ICL after SFT|94.19|64.78
> |Unibias|**94.72**|**65.88**
>
> **Impact of SFT on Biased LLM Components**: Algorithmically, the processes for identifying and mitigating biases in both SFT and non-SFT models are identical. However, we observed a reduction in the coefficients of biased FFN vectors and the magnitudes of label logits of biased attention heads post-SFT in some cases, suggesting that the SFT process may suppress the biased components in LLMs.
>
> ---
> **[Q2]**: Can grid search process be automated?
>
> **[A4]**: Yes, the grid search process in our method is automated. We will release our code to facilitate reproduction and further exploration.
>
> ---
> We are encouraged to find that these experiments and enhancements, prompted by your valuable insights, have substantially strengthened our paper. We look forward to hearing your feedback!

---

> > ### Author Response · Authors · 2024-08-08
> > **References**
> >
> > The references in the rebuttal are as follows:
> >
> > [1] Mitigating Label Biases for In-context Learning. ACL 2023.
> >
> > [2] Prototypical Calibration for Few-shot Learning of Language Models. ICLR 2023
> >
> > [3] Large Language Models Are Not Robust Multiple-Choice Selectors. ICLR 2023
> >
> > [4] Large Language Models Sensitivity to The Order of Options in Multiple-Choice Questions. Findings of NAACL 2024.

---

### Author Rebuttal · Authors · 2024-08-07

Dear Reviewers,

We would like to express our gratitude for the time and effort you've dedicated to reviewing our paper.

We are deeply grateful for your recognition of our work:
* The proposed UniBias method is novel (*Reviewer BhZE, 8E5c*), well motivated (*Reviewer LH5i*), straightforward and does not introduce any additional inference time cost (*Reviewer CnNd*), and has potential to become a leading technique in LLM bias mitigation (*Reviewer BhZE*).
* The exploration on internal mechanisms of underlying biases in LLMs is a relatively unexplored area (*Reviewer CnNd*) and provides insights that uncover the root causes of these biases (*Reviewer BhZE*).
* The methodologies and findings are insightful (*Reviewer 8E5c*).
* Experiments are extensive (*Reviewer 8E5c*), demonstrating significant improvements over standard ICL and SOTA calibration methods (*Reviewer BhZE*).
* A critical problem in ICL is investigated in this paper (*Reviewer LH5i*) and the work represents a unique contribution to bias reduction within LLMs (*Reviewer BhZE*).
* The paper is well-written (*Reviewer 8E5c*).

---
In response to your valuable feedback, we have carefully responded to each comment and wish to outline the main changes as follows:
* **Expanded LLM Evaluation**: We conduct additional experiments to evaluate UniBias on more LLMs, including *GPT-J (6B)* and *GPT2-XL (1.5B)*. Detailed results can be found in *Table 1* of the attached one-page PDF.
* **Exploration of Common Biased Components Across Tasks**: We identify and eliminate the common biased components within LLMs and evaluate its performance on multiple tasks. Results are detailed in *Table 4* of the attached PDF. This experiment demonstrates the potential of our work in stimulating diverse bias mitigation methods in a novel perspective of LLM inner structure manipulation.
* **Enhancement in Grid Search Process**: We optimize the grid search process used in our method by providing alternatives of implementing one-time grid search and using unlabeled samples in place of labeled ones. The results of these optimizations are depicted in *Table 2* and *Figure 1* of the rebuttal PDF.
* **In-Depth Analysis**: We further analyze why mitigating LLMs' bias towards labels can alleviate prompt brittleness in our method. We also visualize and analyze the distribution of identified biased attention heads, as shown in *Figure 2* in the PDF.

---
Moreover, We would like to further highlight the contributions of our work.
* **A New Insight for LLM Bias Mitigation**: Unlike existing works based on *external* adjustments of LLM outputs, we mitigate LLM bias through manipulation of LLM *internal* structure. This novel perspective could potentially stimulate future research on LLM bias mitigation from inner structure manipulation, offering a new direction for the field.
* **Unveil Internal Mechanisms of LLM Bias**: We conduct a thorough exploration of the internal mechanisms underlying biases in LLMs, providing deep insights that go beyond superficial observations, revealing the inner causes of these biases.
* **Extensive Evaluation**: UniBias is evaluate across 12 datasets and 4 LLMs, consistently demonstrating its effectiveness.
* **Addressing Prompt Brittleness**: ur experimental results show that our method effectively mitigates prompt brittleness, a critical issue in ICL.
---

In light of your valuable feedback, we find our work has been significantly enhanced during the rebuttal. Again, we would like thank you for your dedication throughout the review process.

Details of our rebuttal are outlined in our individual responses and the attached rebuttal PDF. Please do not hesitate to contact us for any further suggestions or discussions.

With Gratitude,

Authors of Paper 12087

---

> ### Author Response · Authors · 2024-08-14
> **Looking Forward to Your Valuable Response**
>
> Dear Reviewers,
>
> We greatly appreciate your thorough review, insightful feedback, and recognition of our work. In response to your valuable comments, we have diligently provided detailed explanations and conducted additional experiments to address each point raised. We are eagerly looking forward to hearing your valuable feedback!
>
> We would also like to take this opportunity to highlight the contributions of our work:
> - **A New Direction for LLM Bias Mitigation**: Unlike existing works based on external adjustments of LLM outputs, we mitigate LLM bias through manipulation of LLM internal structure. This novel perspective could potentially stimulate future research on LLM bias mitigation from an internal perspective, offering a new direction for the field.
> - **Addressing an Important Issue of Prompt Brittleness**: Our experimental results show that our method effectively mitigates prompt brittleness, a critical issue in ICL.
> - **Unveiling Internal Mechanisms of LLM Bias**: We conduct a thorough exploration of the internal mechanisms underlying biases in LLMs, providing deep insights that go beyond superficial observations, revealing the inner mechanisms of these biases.
> - **Extensive Evaluation**: UniBias is evaluated across 12 datasets and 4 LLMs, consistently demonstrating its effectiveness.
>
> Thank you again for your time and efforts dedicated to reviewing our paper. We welcome any further questions and are very happy to discuss them.
>
> With Gratitude,
>
> Authors of Paper 12087

---

### Decision · Program_Chairs · 2024-09-25

**Decision:**

Accept (poster)

**Comment:**

This paper analyzes the effect of internal components in transformers (e.g., FFs) on the resulting model biases. It also presents a method for mitigating these biases. Reviewers found the problem important, the contribution unique, the method novel and of great potential, and the experiments extensive. Concerns regarding experiments and the link between the claims and the experiments are mostly resolved.